# Fungal Diseases in Elasmobranchs and Their Possible Treatment with a Special Mention to Azole Antifungal Agents

**DOI:** 10.3390/ani14010043

**Published:** 2023-12-21

**Authors:** Daniela Cañizares-Cooz, Daniel García-Párraga, Emma Plá-González, Carlos Rojo-Solis, Teresa Encinas, Pablo Morón-Elorza

**Affiliations:** 1Department of Pharmacology and Toxicology, Faculty of Veterinary Medicine, Complutense University of Madrid, Av. Puerta de Hierro s/n, 28040 Madrid, Spain; tencinas@vet.ucm.es (T.E.); p-moron@hotmail.com (P.M.-E.); 2Fundación Oceanogràfic de la Comunitat Valenciana, C/Eduardo Primo Yúfera (Científic) 1B, 46013 Valencia, Spain; dgarcia@oceanografic.org (D.G.-P.); epla@oceanografic.org (E.P.-G.); 3Veterinary Services, Oceanogràfic, Ciudad de las Artes y las Ciencias, C/Eduardo Primo Yúfera (Científic) 1B, 46013 Valencia, Spain; crojo@oceanografic.org

**Keywords:** mycosis, sharks, rays, fusarium, azolic antifungals

## Abstract

**Simple Summary:**

Fungal diseases, despite their low incidence in fish, appear to be lethal in many teleost and elasmobranch species. There are many fungi involved, including *Paecilomyces* spp., *Exophiala* spp. and *Fusarium* spp., among the most frequently diagnosed. Fungal diseases in elasmobranchs have been documented in animals under human care in aquariums since 1980. Most cases have resulted in high mortality due to a lack of information concerning the treatment and medical and environmental management of mycoses in these species. Antifungal drugs are frequently prescribed to improve clinical signs or lesions in animals affected by fungal infections. Azole agents appear to be the most effective antifungals in the treatment of systemic fungal infections. However, more evidence concerning the use of these drugs in elasmobranchs is needed.

**Abstract:**

Introduction: Elasmobranchs currently constitute an important part of the animal collection of many aquariums worldwide. Their maintenance under human care has allowed us to describe and identify new pathogens and diseases affecting them, as well as to determine different treatments for these diseases. Great advances in elasmobranch husbandry have been developed. Methods: A search was performed on scientific databases as PubMed and other specialized sources (IAAAM archive). Results: Little information on pharmacotherapeutics is available in this taxonomic group, and treatments lack a scientific base and instead are frequently dependent on empirical knowledge. Pharmacokinetic studies are the first step to determining therapeutic protocols that are safe and effective. The available bibliography shows that a majority of the mycoses recorded in cartilaginous fish are severe, aggravated by the fact that the antifungal treatments administered, following the guidelines used for teleost species, are ineffective in elasmobranchs. Azoles appear to be a promising group of antifungals for use in treating systemic mycoses in sharks and rays. Conclusions: Based on the findings of this review, it is essential to investigate the pharmacokinetics of the different antifungals in these species in order to provide therapeutic options for fungal infections in cartilaginous fish.

## 1. Introduction

Elasmobranchs are an important part of the species collections of many aquariums around the world. Over half of the aquariums in Europe maintain different specimens belonging to this group of emblematic animals [1]. The management of these species under human care has allowed for the description of their diseases and their possible treatments. However, there is little information available on pharmacotherapeutics in elasmobranchs. According to the most extensive pathological review in elasmobranch cases (from 2013), infectious and inflammatory processes are the most prevalent diseases in cartilaginous fish (33.5%). Fungal infections represent only 0.6% of all the diseases registered in this data set of animals maintained under human care [2]. Although fungal infections are uncommon in elasmobranchs, affected cases have a high mortality rate and some authors believe there are emerging diseases [3].

Fungal infections are frequently described in teleosts. Within the fungal disease complex of fish, diseases caused by fungal-like organisms and actual fungal infections are differentiated [4]; this is because some microorganisms that are very similar to fungi in their morphology were considered fungi in the past, but have been reclassified by recent phylogenetic studies. It is important to emphasize this gap, because real fungal infections are less common, have higher mortality rates and require a different pharmacotherapeutic approach [2].

Fungal diseases in elasmobranchs are predominantly caused by *Fusarium* spp., a large group of pathogens that are part of the Fusarium Solani Species Complex (FSSC) [5]. Due to the effectiveness of azole antifungal drugs, specifically voriconazole, these drugs might be an option for the treatment of mycosis in elasmobranchs [6]. An updated review of the most frequent fungal diseases in elasmobranchs and the efficacy of azole antifungal drugs is presented in this article.

## 2. Fungal-like Infections in Fish

Some diseases affecting fish included within this group are: Saprolegniasis, Branchiomycosis, Epizootic Ulcerative Syndrome (EUS) and Ichthyophonosis [7]. The first three diseases are caused by microorganisms of the Oomycetes class, which were classified as fungi for many years, but recent phylogenetic studies have demonstrated that they are closely related to diatoms, golden algae and brown algae. These oomycete microorganisms are also known as water molds, and they have a fungal-like filamentous growth that makes them similar to fungal organisms, but they are protists, and their cellular membrane is made of cellulose instead of chitin. Mold diseases commonly affect freshwater fish because seawater is not tolerated by these microorganisms. The fact that 95% of sharks and rays are marine species makes them less susceptible to these diseases [8,9]. On the other hand, *Ichthyophonus hoferi* was also described as a fungus, but currently has been classified into the Mesomycetozoea class; however, it should be taken into account that it has a larger host spectrum and has been documented anecdotally in elasmobranchs.

## 3. Real Fungal Infections Affecting Fish

Real fungal diseases in fish are uncommon and are typically identified in immunosuppressed animals. These infections are most frequently caused by Microsporidians, *Fusarium* spp., *Paecilomyces* spp. and *Exophiala* spp. [3].

### 3.1. Predominant Fungal Diseases

#### 3.1.1. *Microsporidium* spp.

Microsporidians were classified as protozoa for a long time, but recently have been relocated to the Fungi kingdom because of their phylogenetic relationships with Zygomycetes [10]. These microorganisms are intracellular pathogens that can infect a large number of animal species, including fish. The infection of the host cell causes the formation of a cyst called a xenoma. Microsporidians are more common in freshwater teleosts, but they can also infect seawater fish species [11]. One case of microsporidiosis has been reported in elasmobranchs. In 2008, a common stingray (*Dasyatis pastinaca*) was captured and several tumor-like lesions were observed. The pathological analysis revealed that the tumors were cysts full of fungal spores, which resulted in the discovery of a new microsporidian species called *Dasyatispora levantinae*, deferring its lesions from the teleost’s microsporidian xenomas [12].

#### 3.1.2. *Exophiala* spp.

The genus *Exophiala* spp. includes many fungi that can be found in the soil and water of marine ecosystems. The most common pathogen species for fishes are *E. angulospora*, *E. pisciphila* and *E. salmonis*. The infection starts with a wound which is colonized by the fungus. This develops into an ulcer that invades the deep tissues [13]. This fungi genus is also implicated in Phaeohyphomycosis, a rare mycotic infection caused by a heterogeneous group of dematiaceous and filamentous fungi that affect the skin and subcutaneous tissues, and occasionally results in severe systemic infections [14]. These processes have been documented not only in fish but also in humans [15]. *Exophiala* infections are more frequent in fish farms, affecting salmonids, demersal teleost and syngnathids, being anecdotic in elasmobranchs [11,16].

#### 3.1.3. *Fusarium* spp.

*Fusarium* spp. is a saprophyte genus that is ubiquitous in the environment and can be isolated from soil and plants. *Fusarium solani* is the most frequent species involved in infections, producing a necrotizing granulomatous dermatitis. Fusariosis is a disease that affects seawater species and is uncommonly found in freshwater fish [17,18]. The Fusarium Solani Species Complex (FSSC) has multiple phylogenetic groups; it is known that FSSC2 (*Fusarium keratoplasticum*) and FSSC9 (*Fusarium* haplotype 9X) are the most frequently implicated in fungal infections in elasmobranchs. FSSC2 is also frequently isolated in human mycosis and is therefore a zoonotic risk [5].

Despite the absence of information in free-range fish, there are some reports of *Fusarium* spp. infections in wild marine animals [19,20]. Marine elasmobranchs are very susceptible to *Fusarium solani* mycosis and are responsible for the “Bonnethead shark disease” that appears in animals of the *Sphyrna* genus under human care. Most documented cases result in high mortality rates, despite treatment [3]. This disease can impact elasmobranch conservation programs in aquariums and marine recovery centers, as the eight *Sphyrna* species are currently classified as “endangered” and five of them are considered “critically endangered” according to the International Union for Conservation of Nature (IUCN) [21].

A list of the reported *Fusarium* spp. Infections in marine species is summarized in Table 1.

### 3.2. Evolution and Magnitude of Fungal Diseases in Elasmobranchs

The first case of a fungal disease in elasmobranchs was documented in 1986; it was caused by *Exophiala pisciphila*, resulting in the death of a 2-month-old smooth dogfish (*Mustelus canis*) in the New York Aquarium. The fungi was isolated from the skin and brain tissues of the fish [37].

A second case of a fungal infection was reported in elasmobranchs in 1989; it affected young bonnethead sharks (*Sphyrna tiburo*) born at the National Aquarium in Baltimore, Maryland. All animals died shortly after birth and presented with lethargic and stuporous behavior. During the necropsy, ulcerated skin lesions were observed, along with associated massive muscle necrosis—some of it extending to the cartilage. The histopathological examination revealed the presence of hyphae in all the injured tissues, in addition to a focus extended to the kidney in one of the animals. After culturing the collected samples, it was discovered that the etiology of these infections was *Fusarium solani*, associated in one of the infected ulcerated lesions with *Pseudomonas putrefaciens* [38].

Six years later, another fungal infection was documented in two hammerhead sharks (*Sphyrna lewini*) kept under human care at the Waikiki Aquarium (Hawaii). In addition to changes observed in their behavior—abnormal swimming and anorexia—inflammatory skin lesions were located along the lateral line and cranial pores showed a viscous whitish secretion. A granulomatous exudative fungal dermatitis was diagnosed using histopathology, and after the euthanasia of both animals, a microbiological study of the lesions and exudate revealed the presence of *Fusarium solani* and two bacteria (*Vibrio damselae* and *Vibrio alginolyticcs*) [26].

New fungi species were identified in 2011 as causative agents of fungal disease in sharks. The first fungal infection in elasmobranchs caused by Paecilomyces lilacinus resulted in the death of a great hammerhead shark (Sphyrna mokarran) in less than 2 months. The animal began to have alterations in swimming, which caused external lesions on the skin and eyes. Despite the administration of treatment (amikacin 5 mg/kg i.m., prednisolone 5 mg/kg i.m. i.m., ceftazidime 22 mg/kg i.m. i.m. and vitamin supplementation), the animal died, and the necropsy revealed multiple ulcerations on the skin (along the nasal fossae, anterior margin of the cephalofoil, midventral pectoral girdle and ventral aspect of the caudal fin), bilateral corneal perforation and hemorrhagic gills. Internally, cloudy hemorrhagic free fluid was found inside the celomic cavity, along with a friable and dark liver, pale areas in the spleen and parasites in the area of the spiral valve. Microscopic examination identified the presence of hyphae in the liver, myocardium and gills. Samples from the spinal fluid, kidney, spleen and coelomic cavity were cultured, and colonies of Paecilomyces lilacinus, *Vibrio* spp. and alpha-hemolytic streptococci were isolated [39].

The other new fungi described in elasmobranchs are *Exophiala* spp. and Mucor circinelloides, isolated in the first reported case of mycosis in a zebra shark (Stegostoma fasciatum) kept under human care. The animal initially presented with swimming difficulties that did not improve with treatment (diazepam 0.58 mg/kg i.m. i.m., chloramphenicol 30 mg/kg i.m. every 2 days and ceftazidime 22 mg/kg i.m. every 4 days; prednisolone 5 mg/kg half i.m. i.m./hald i.v. in addition to vitamin supplementation). The analytical changes described included an increase in immature red blood cells and a progressive decrease in the total leukocyte count throughout the pathological process. Humane euthanasia was finally performed due to clinical deterioration. The necropsy revealed that the animal was cachectic, its gills were pale and covered with mucus and there were no skin lesions. However, foci of necrosis in the spleen and liver were found. In addition, hyphae were detected in the myocardial tissue and liver lesions. Microscopic observation of the tissues found two different morphologies of hyphae in the hepatic samples. Two different fungal colonies were isolated from the liver—one of them identified as *Exophiala* spp. and the other Mucor circinelloides, being the first case described for this genus. In addition, bacterial cultures revealed the presence of *Enterococcus* spp. [39]. In the last decade, more fusariosis cases have been documented in *Sphyrna* genus sharks kept under human care in aquariums [6,27]. The first *Fusarium* infection in a batoid was reported in 2015 in a black stingray (*Taeniura melanopsila*), which presented with ulcerated skin lesions in the ventral zone. The animal died one week after the appearance of the lesions. The necropsy and the culture of the tissue samples determined that the etiology of the disease was *Fusarium solani* [30].

A review of the literature available revealed that the most frequent etiological agents of fungal diseases in elasmobranchs are *Fusarium solani*, *Exophiala* spp., *Paecilomyces lilacinus* and *Mucor circinelloides*. In these mycoses, skin lesions are initially detected, involving necrotizing foci of fungal growth that affect internal tissues, resulting in systemic mycosis in some cases. In all cases, the fungi were isolated from the lesions, along with opportunistic bacteria that colonized the damaged tissues. The presence of bacteria aggravated the clinical signs in most cases and required the addition of antibiotic therapy to the medical treatment.

#### 3.2.1. Diagnosis of Fungal Diseases in Elasmobranchs

Affected tissue samples are required for the diagnosis of fungal diseases. Histopathology and hyphae morphology of the lesions can be useful for the presumptive diagnosis, but culture is required for the definitive identification of the fungus. Inoculation of the samples can be performed in culture plates or slats, requiring incubation at room temperature [4,39]. The most common fungal media are Sabauraud agar and potato dextrose agar. Antimicrobials may be added to the media in order to decrease the competitive growth of bacteria. In addition, for specie identification, PCR and sequencing is required [11].

#### 3.2.2. Prognosis and Treatment of Fungal Diseases in Elasmobranchs

The prognosis of fungal infections is typically poor in elasmobranchs and indeed most cases are fatal despite the administration of multimodal therapies that include analgesics, antibiotics, antifungal drugs and changes in the water parameters [27]. In recent years, antifungal drugs have been increasingly administered in elasmobranchs to fight fungal infections, typically combined with other husbandry strategies that involve changes in water parameters; however, there is still no scientific evidence for their efficiency.

To date, the effectiveness of antifungal therapies in elasmobranchs is unknown. Most of the documented cases of fungal infection have resulted in massive mortalities despite treatment, and just three cases have had a positive outcome reporting survival in the affected animals [6,27,31]. Based on the available data, it can be suggested that the treatment of fungal infections in elasmobranchs requires the following management to maximize recovery and the chances of survival:

Changes in the water parameters, including salinity, oxygen saturation and temperature. It should be noted that increasing water temperature slows fungal growth, but it can also benefit the survival of some pathogenic bacteria [6,27].Antibiotic drugs are useful for treating secondary infections. Fluoroquinolones are the most frequently used antimicrobial group in these cases [27,31].Antifungal therapy; azole antifungals seem to be the most effective drugs for mycosis treatment in fish according to empirical knowledge, although the dose needed to produce a therapeutic effect is unknown in many species, including elasmobranchs. Terbinafine has been used topically and orally in addition to azole antifungal drugs. Voriconazole appears to be the best choice in all reported cases because of its Minimal Inhibitory Concentration (MIC) against *Fusarium solani*, but the wide dosage range used (4 mg/kg p.o.; p.o. 12 mg/kg p.o. and 50 mg/kg p.o.) makes it difficult to determine which dose is optimal. This lack of scientific information regarding the drug concentration achieved in plasma and other tissues justifies the review carried out in this article and highlights the need for pharmacokinetic and pharmacodynamic studies in this species [6,27,31].

A list of the antifungals used in elasmobranches is included in Table 2.

## 4. Azole Antifungal Drugs as Treatment of Fungal Diseases in Elasmobranchs

Azoles are a group of antifungal drugs that are chemically characterized by an imidazole ring, which contains two or three nitrogen atoms [40]. Their mechanism of action is based on the inhibition of the 14α-demethylase of the fungus, a microsomal enzyme that belongs to the cytochrome P450 family (CYP459) [41]. The 14α-demethylase regulates lanosterol conversion into 4,4-dimethylcholesta-8,14,24-trienol, which is one of the metabolic rote steps for ergosterol synthesis. Ergosterol is a sterol that is part of the plasmatic membrane of fungal cells and has a fundamental role in the membrane’s stability, affecting its fluidity and permeability [42]. Azoles inhibit this metabolic pathway, blocking ergosterol production, which causes the activation of secondary enzymes that convert lanosterol into unusual metabolites that are toxic to the fungus. As a consequence of its primary pharmacological activity, a fungistatic effect is produced, caused by the difficulty of the fungus to grow due to alterations in the integrity and functionality of the plasmatic membrane of the cell and its organelles—among them the mitochondria—resulting in an accumulation of reactive oxygen species (ROS) [43]. Furthermore, it has recently been shown that some drugs in this group also have fungicidal properties, possibly related to the toxic effect of the molecules resulting from the alternative metabolic pathways of lanosterol [44].

Azole drugs can be divided into three two groups: imidazoles and triazoles, depending on the number of nitrogen atoms contained in the imidazole rings [40]. Miconazole was the first imidazole derivate used by clinicians, and it is still available for topical treatment [45]. The triazole class was developed later, with a similar chemical structure as the imidazole group, but with a nitrogen atom added into the imidazole rings. Itraconazole and fluconazole were the first-generation triazoles, which showed high levels of efficacy and a larger spectrum compared with amphotericin B and imidazoles. Ten years later, in 2002, second-generation triazoles were developed (voriconazole, posaconazole, ravuconazole, albaconazole, isavuconazole, efinazonazole) [46]. Despite sharing the same mechanism of action with imidazoles, triazoles have fewer secondary effects and are available for oral treatment as well as intravenously for hospital use [45]. Other azole derivate molecules have been studied as antifungal agents (oxazoles, thiazoles, tetrazoles, carbazoles), due to increases in antimicrobial drug resistance, being an interesting treatment alternative for the future [47].

Triazoles are metabolized in the liver by different cytochrome P450 complex enzymes. For example, voriconazole is metabolized by the CYP2C19, CYP2C9 and CYP3A4 enzymes, and has a moderate inhibitory effect on CYP2C8/9, CYP2C19 and CYP3A4 in humans. For this reason, these drugs can produce hepatic toxicity [48]. In addition to hepatotoxicity, azole drugs have other adverse effects related to long-term administration regimes, as ergosterol is a starting component of mammal steroid synthesis; as such, the inhibition of these pathways can cause deficiencies in some hormones. Phototoxicity, adrenal insufficiency and neuropathies are some of the adverse effects of the long-term use of azole drugs in humans [49].

One of the effects of the use of antifungal drugs in both human and veterinary medicine is the presence of resulting residues in the environment [50]. This may result in the exposure of many ubiquitous pathogenic fungi to these drug residues, generating antifungal resistance [41]. Despite this, to date, no resistance to these drugs has been observed in susceptibility tests performed with fungi isolated from elasmobranchs [31].

In veterinary medicine, the most common azole drugs used for fungal infection treatment are itraconazole and voriconazole. In dogs, it has been reported that after an oral dose of voriconazole of 6 mg/kg, the drug reaches maximum plasma concentrations (C_max_) of 3.07 µg/mL [51]. Oral doses of voriconazole (4 to 6 mg/kg) have been used in cats, at a concentration of 2.2 µg/mL, and no adverse effects were observed [52]. The terminal elimination half-life is significantly shorter in dogs (3.13 h) compared to cats (40.50 h). However, detectable plasma concentrations have been found after 24 h in both species [51,52]. In belugas (*Delphinapterus leucas leucas*) administered with voriconazole at doses of 3 mg/kg twice a day, plasma concentrations of 2 µg/mL can be reached, potentially proving effective for the treatment of reported *Fusarium solani* infections [34].

In birds, the use of azoles has been more frequently studied; in Humboldt penguins (*Spheniscus humboldti*) oral itraconazole at 20 mg/kg reaches plasma concentrations of 1.4 µg/mL, being above 0.5 µg/mL, which is the minimum concentration considered effective in humans [53,54], but with a short terminal elimination half-life (5.06 h). Voriconazole has been tested in African penguins (*Spheniscus demersus*) orally at 5 mg/kg, obtaining a C_max_ of 1.89 µg/mL, which is very close to the minimum effective concentration in humans (2 µg/mL) and other mammal species. The elimination phase is longer than for itraconazole (10.92 h) [55,56].

In reptiles, voriconazole has been prescribed in some aquatic species such as the green sea turtle (*Chelonia mydas*), using a dose of 10 mg/kg without observing any adverse effects [57]. A pharmacokinetic study in red-eared sliders (*Trachemys scripta elegans*) revealed that a single 10 mg/kg dose of subcutaneous voriconazole produced plasma concentrations of over 1 µg/mL during the first 24 h, without adverse effects [58,59].

As previously mentioned in this review, the use and clinical evidence of azoles in fish is limited, mainly due to the lack of scientific information available on their pharmacokinetics and pharmacodynamics. A single pharmacokinetic-toxicity study with miconazole in cyprinids (*Labeo rohita*) was found during our search, in which no adverse effects were observed after oral administration of 25.22 mg/kg, reaching a C_max_ of 20.28 µg/mL with a prolonged elimination half-life (77 h) [60].

Although azoles have been used empirically for the treatment of fungal diseases in elasmobranchs, there are no studies that show their kinetic disposition and efficacy. Voriconazole has produced one of the lowest MIC against *Fusarium solani* (3 to 8 µg/mL) during in vitro studies as well as against other fungi isolated from sharks such as *P. lilacinus* (0.5 µg/mL MIC) [31,61].

In 2017, a hammerhead shark (*Sphyrna tiburo*) and a bonnethead shark (*Sphyrna lewini*) survived a fungal skin infection caused by *Fusarium solani*. Pharmacological treatment was established after the isolation of the fungus. Previous in vitro inhibition studies recommended voriconazole as the drug of choice in elasmobranches because of its MIC_90_ of 2 µg/mL [62], although doses in sharks range from 5 mg/kg p.o to 50 mg/kg p.o b.i.d. Changes in environmental parameters combined with pharmacological treatment may include modification of the water temperature to 29.5 °C [6].

A year later, another fusariosis case in hammerhead sharks (*Sphyrna lewini*) was published, in which four of the fourteen affected animals finally survived. The treatment prescribed included changes in water conditions consisting of a salinity (from 36.5 to 36.9 g/L) and temperature increase (from 27 to 29 °C) and the maintenance of water oxygen saturation between 105 and 120%. In addition to environmental changes, pharmacological treatment was initially performed with terbinafine (10 mg/kg p.o.,), followed by 12 mg/kg 3 times/week), enrofloxacin (10 mg/kg 3 times a week) and cefovecin (8 mg/kg every 14 days) [27].

The most recently reported *Fusarium solani* infection was in 2021 and involved a roughtail stingray (*Dasyatis centroura*). Skin lesions suggestive of fusariosis were observed all over the body and samples collected from the lesions confirmed the pathogen. The treatment was based on topical terbinafine (Lamisil 1% cream) and itraconazole (5 mg/kg p.o. s.i.d.) After a week of antifungal therapy, itraconazole was replaced by voriconazole (4 mg/kg p.o. s.i.d) and enrofloxacin (10 mg/kg p.o. s.i.d.) was added to prevent secondary infections. Three months after starting treatment, the skin lesions began to regress, and fewer hyphae were seen in the tissue samples, ultimately resulting in the survival of the animal from the infection [31].

Sporadic analyses of voriconazole plasma concentrations in sharks have revealed some important information: voriconazole administered orally at a dose of 30 mg/kg achieves plasma concentrations of 1.65 µg/mL in bonnethead sharks (*Sphyrna tiburo*) and oral administration of voriconazole administered at 50 mg/kg in hammerhead sharks (*Sphyrna lewini*), produces plasma concentrations of 1.2 µg/mL [6].

Recent studies have facilitated the development of pharmacokinetics studies in elasmobranchs, creating a methodology that can be useful for studying other drugs such as azoles. These studies have described the great interindividual variabilities and low absorption of some drugs when administrated orally to elasmobranchs, and support alternative administration routes such as intramuscular administration instead [63,64,65]. Voriconazole has multiple pharmaceutical formulations: tablets (50 mg, 100 mg, 200 mg) and suspension (40 mg/mL) for its oral administration, and intravenous solution (10 mg/mL solution and 200 mg vial) for infusion [66,67]. All current voriconazole pharmaceutical formulations are approved for human medicine, so veterinarians are forced to prescribe these under their own judgement and responsibility. Intravenous voriconazole formulations have been used in animals intramuscularly and subcutaneously without producing adverse effects [57,58,59,67]. These studies support the administration of voriconazole via these routes in different animals. Due to the extremely high oral doses that seem to be required in sharks (30–50 mg/kg), parenteral administration could be a more efficient route in elasmobranchs, allowing a reduction in the dosages needed while guaranteeing high plasma concentrations.

The clinical use of voriconazole in elasmobranchs, although still limited, is promising as an effective treatment against fungal infections Figure 1. However, further pharmacokinetic, pharmacodynamic and toxicity studies are needed to be able to assess the use of azoles in these species.

## 5. Conclusions

The information available concerning the treatment of fungal diseases in elasmobranchs is limited and is based on practical experience, but does not have a scientific base. As fungal diseases frequently result in high mortality rates in elasmobranchs and can compromise the health of these animal species, more research is needed. According to the literature summarised in this review, the use of azole antifungal drugs, specifically voriconazole, can be a promising option for the treatment of mycoses in sharks and rays. More studies on the pharmacokinetics and efficacy of this group of drugs in fish (including elasmobranchs) could be useful for proposing a safe and effective posology regime for dealing with fungal infections.

## Figures and Tables

**Figure 1 animals-14-00043-f001:**
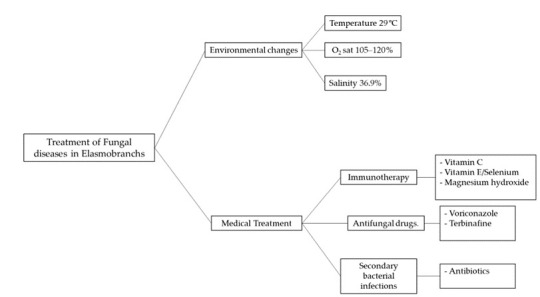
Multimodal therapy for the treatment of Fungal diseases in elasmobranchs.

**Table 1 animals-14-00043-t001:** Reported *Fusarium* spp. infections in marine species.

Group	Species	Situation	References
Invertebrates	*Penaeus japonicus*, *P. monodon*, *P. vannamei*, *P. californiensis*, *P. stylirostris*, and *P. merguensis*	Under Human Care	[22,23]
Teleosts	*Pomacanthus paru*, *Pomacanthus arcuatus*, *Holocanthus bermudensis*, *Holacanthus ciliaris*, *Scarus coelestinus*, *Scarus guacamaia*, *Hippocampus erectus*	Under Human Care	[24,25]
Elasmobranchs	*Sphyrna lewini*, *Sphyrna tiburo*, *Heterodontus portusjacksoni*, *Dasyatis centroura*, *Taeniura melanopsila*	Under Human Care	[6,26,27,28,29,30,31]
Reptiles	*Lepidochelys kempii*, *Caretta caretta*	Under Human Care and Wild	[19,20,32]
Mammals	*Pseudorca crassidens*, *Delphinapterus leucas leucas*, *Lagenorhynchus acutus*, *Kogia breviceps*, *Phoca vitulina*, *Halichoerus grypus*, *Zalophus cafornmnus*	Under Human Care	[33,34,35,36]

**Table 2 animals-14-00043-t002:** Administered antifungal drugs in elasmobranch species to treat fungal infections.

Antifungal Drug	Patient Species	Etiology	Posology	Coadministration	Comments	Reference
Itraconazole	*Dasyatis centroura*	*Fusarium solani*	5 mg/kg; p.o. p.o.; s.i.d s.i.d.	Terbinafine	Not effective, changed to voriconazole	[31]
	*Heterodontus portusjacksoni*		10 mg/kg p.o. p.o.; every 2 days	Cefazidime Enrofloxacin	100% mortality	[28]
Terbinafine	*Sphyrna lewini*		10 g/kg; p.o. p.o.	Voriconazole	Low % of remission	[27]
	*Dasyatis centroura*		5 mg; topical	Itraconazole Voriconazole Enrofloxacin	Not seems to be effective without systemic antifungal treatment	[31]
	*Heterodontus portusjacksoni*		20 mg/kg p.o. p.o.; s.i.d s.i.d.; 40 mg/kg p.o. p.o.; s.i.d s.i.d.	Cefazidime	100% mortality	[29]
Voriconazole	*Sphyrna tiburo*		30 mg/kg; p.o. p.o.; s.i.d s.i.d.	-	Recovery	[6]
	*Sphyrna lewini*		50 mg/kg; p.o. p.o.; b.i.d b.i.d.	-	Recovery	
	*Sphyrna lewini*		12 mg/kg; p.o. p.o.	Terbinafine Marbofloxacin Enrofloxacin Cefovecin Ceftadizime	High mortalities due to sepsis	[27]
	*Dasyatis centroura*		3–4 mg/kg; p.o. p.o.; s.i.d s.i.d.	Terbinafine Enrofloxacin	Recovery	[31]

## Data Availability

The data presented in this study are available on request from the corresponding author.

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
