# Peer review of "Fungal Diseases in Elasmobranchs and Their Possible Treatment with a Special Mention to Azole Antifungal Agents"

_animals, 2023, doi:10.3390/ani14010043_

Round 1

Reviewer 1 Report

Comments and Suggestions for Authors

Overall, this is a well written and organized review paper. The authors reviewed and summarized the literature evidence involving categorization and diagnosis of fungal and fungal-like infectious diseases, treatments of azole antifungal drugs and potential applications of this agents on elasmobranchs. This review will provide better understanding of fungal infections on elasmobranchs, especially sharks. It also pointed out the lacking of scientific information in regarding to pharmacokinetics of azole antifungal drugs can hinder the efforts on improving fish health against fungal infectious diseases. I listed few items that may help improve the manuscript as below, and suggest the authors to consider involve them in a revised version.

1.     Citation is required at line 80.

2.     In order to keep the logic coherence and make the text easier to understand, I suggest the authors to remove the bullet marks before the paragraphs at line 148 – 161 and 162 – 171, and add some transition sentences before these two paragraphs.

3.     There is a third type of azole antifungal drug, namely Thiazole, with sulfur atom on imidazole ring. This type of antifungal drug, different from imidazole and triazole-classes, interrupted the composition of fungal membrane (Borelli 2008, Chemotherapy), and may be an alternative approach for marine species treatment.

4.     I wonder if any literature evidence explained or gave possible reasons of extremely high oral dosage on Elasmobranchs treatment but achieved only average blood concentrations of these azole agents. If yes, I suggest include this part in the manuscript.

Author Response

Dear Reviewer.

Thank you very much for your suggestions. We have included the changes in our manuscript.

Kind Regards.

Daniela

Reviewer 2 Report

Comments and Suggestions for Authors

Undoubtedly, fungal diseases in elasmobranchii are an important problem for veterinary medicine and many aquarium fish lovers. Indeed, there are a number of reports on this topic in the literature, but there is a lack of analytical reviews that especially systematize the latest achievements in the field. Thus, the proposed review is relevant, necessary and interesting, therefore, it will be well cited. The text is written well and logically, provided with the necessary links, the text contains generalizing tables. The conclusions of the article are logical and supported by materials presented in the body of the article. I also really like that the authors not only simply state facts from literature, but analyze the facts and express their attitude towards them. This means a high scientific level of review. I recommend accepting this review, however a minor revision is needed to improve it and make it more appealing to readers. I will kindly ask the authors to make one or two figures.

Author Response

(The authors gave the same response as above.)

Reviewer 3 Report

Comments and Suggestions for Authors

The abstract should be structured with an introduction, objectives, methodology, results, and conclusion. In its abstract there is only an introduction without results and methodology, in this sense, I request that the writing of this topic be restructured.

The introduction is well written;

However, I question:

What is the methodology used in this review? What criteria were used in choosing the studies? Was it studies from the last 10 years?

The conclusion needs to be rewritten as it feels like an introduction. What are the main conclusions of this research? What results are relevant for future studies in the area?

Author Response

Dear Reviewer.

Thank you very much for your suggestions. We have included the changes in our manuscript.

We specified the methodology in the abstract as was suggested. The studies chosen are all the literature available about the subject, which is very limited. The main conclusion of the review is the lack of a scientific base for the treatment of fungal diseases in elasmobranchs, and that azole antifungal drugs are a promising option according to the clinical experience. Our review is important because summarises all the information available and leads the next step for the research in the treatment of fungal diseases in elasmobranchs.

Kind Regards.

Daniela

Round 2

Reviewer 3 Report

Comments and Suggestions for Authors

Corrections have been made, I agree with the publication of the manuscript